

# Dynamically-optimal models of atmospheric motion

Alexander G. Voronovich

Physical Sciences Laboratory/NOAA, Boulder CO 80305 USA

*Correspondence to*: Alexander G. Voronovich (alexander.voronovich@noaa.gov)

**Abstract.** A derivation of a dynamical core for the dry atmosphere in the absence of dissipative processes based on the least action (i.e., Hamilton's) principle is presented. This approach can be considered the finite-element method applied to the calculation and minimization of the action. The algorithm possesses the following characteristic features: (1) For a given set of grid points and a given forward operator the algorithm ensures through the minimization of action maximal closeness (in a broad sense) of the evolution of the discrete system to the motion of the continuous atmosphere (a dynamically-optimal algorithm); (2) The grid points

can be irregularly spaced allowing for variable spatial resolution; (3) The spatial resolution can be adjusted locally while executing calculations; (4) By using a set of tetrahedra as finite elements the algorithm ensures a better representation of the topography (piecewise linear rather than staircase); (5) The algorithm automatically calculates the evolution of passive tracers by following the trajectories of the fluid particles, which ensures that all *a priori* required tracer properties are satisfied. For testing purposes, the algorithm is realized in 2D, and a numerical example representing a convection event is presented.

**1 Introduction**

The models simulating atmospheric dynamics are often built by replacing spatial and temporal derivatives in the continuous equations of motion (like conservation of momentum, mass, etc.) by corresponding finite-difference approximations. In finite-volume versions, the discrete approximations of spatial derivatives are also used for calculation of fluxes. There are numerous

ways to proceed based on experience accumulated amongst different disciplines. The approaches which start from the continuous equations of motion and pursued along these lines essentially ignore, however, the fact that that the governing equations representing atmospheric dynamics themselves follow from the least action, or Hamilton's, principle (Eckart, 1960; Salmon, 1983) (LAP for brevity), and it is reasonable to take advantage of this fact. LAP states that the action, which is a time integral of the difference between the total kinetic and potential energy (i.e., the Lagrangian) of the system, is minimal for the actual evolution of

a mechanical system. Application of LAP leads to the Euler-Lagrange equations of motion which include the second order derivatives of the state variables in time. By using the Liouville transformation these equations can be cast into the Hamiltonian equations which, being resolved with respect to the first-order time derivatives, are thus better amenable to both analytical and numerical solutions (Salmon, 1988).

To build a computer model of atmospheric motion, one has to pass from a continuous to a discrete system. There are at least two

ways to achieve this within Lagrangian/Hamiltonian approach. The first one is to replace a continuous Hamiltonian or Lagrangian by a discrete analogue (Salmon, 1983). If the Hamiltonian equations are formulated in non-canonical coordinates (e.g. Rolstone and Bruce, 1995), one has to approximate also the Poisson bracket (Eldred et al, 2019; Salmon, 2004). By maintaining corresponding symmetries, one tries to ensure also that the conservation laws of the original equations are inherited by the discretized ones as well (Salmon, 2004). An important advantage of the Hamiltonian and Lagrangian descriptions is that different

approximations (like quasi-hydrostatic) can be done within this approach by modifying corresponding Hamiltonian or Lagrangian (Shutts, 1989; Salmon and Simth, 1994; Rolstone and Bruce, 1995).



The second way is after selection of a discrete set of parameters to approximate an action of the continuous atmosphere. To do this one has also to choose an observation (or forward) operator, i.e. a mode of interpolation allowing to unambiguously calculate state of the atmosphere at any spatial point based on the set of discrete parameters. Using this operator, one approximately calculates

action density of the continuous atmosphere in terms of the discrete parameters, integrates it over the space, and by minimizing the result obtains ordinary differential equations (ODE) governing their evolution. Such an approach seems to be more consistent. From its standpoint replacement of the continuous Hamiltonian/Lagrangian by a discrete analog can be considered as a piecewise-constant spatial approximation of corresponding density. One can, however, use more accurate approximations; in this work, in particular, a piecewise-linear approximation is applied instead. In this case action density becomes piecewise continuous. One

can refer to such atmospheric models as "dynamically-optimal", since for a given set of discrete parameters and a given forward operator the governing ODE follow unambiguously ensuring minimal action and thus the best approximation (in a broad sense) of the dynamic of a continuous atmosphere. Such an approach is essentially a combination of finite-element method and LAP.

The approach based on approximation of continuous action was considered recently in (Gawlik and Gay-Balmaz, 2021). In this work, however, an action of the compressible atmosphere was calculated in a noncanonical coordinates, what leads to a

minimization of action under certain constraints on variations. This interesting technique differs significantly from the approach pursued in this paper.

We are considering in this work the approximation of the action in spatial domain only assuming continuous dependence on time. However, one can discretize calculation of action not only in spatial but in time domain as well. Corresponding discrete time-evolution schemes are called variational integrators (Mardsen and West, 2001; Lew et al, 2003). Such an approach allows also to

use different time steps in different areas (Asynchronous Variational Integrators), an option which should be of interest to the development of dynamical cores for weather- and climate prediction models.

## 2 The action for a continuous atmosphere

This section describes mostly standard transformations (see e.g. Eckart,1960; Salmon,1988; Shutts,1989; Salmon and Smith, 1994) that precede transition from continuous to discrete forms for the governing equations. We will consider here the case of a dry, rotating atmosphere which is a base of a dynamical core. We begin with the following equations that represent the adiabatic motion of dry air in the absence of losses:

$$\partial_t \vec{v} + \left(\vec{v}\nabla\right)\vec{v} + \nabla\phi + 2\vec{\Omega}\times\vec{v} + \frac{\nabla P}{\rho} = 0,$$

$$\partial_t \rho + \nabla\left(\rho\vec{v}\right)v = 0, \tag{1}$$

$$\partial_t s + \vec{v}\nabla s = 0.$$

Here the usual notations are used with $s$ being entropy per unit mass, $\phi$ the geopotential, and $\vec{\Omega}$ an angular velocity vector. The pressure $P = P\left(\alpha, s\right)$ is considered to be a function of the specific volume $\alpha = 1/\rho$ and entropy (to distinguish pressure from a conjugated momentum $\vec{p}$ to be introduced later, we denoted pressure with capital letter). In the Lagrangian coordinates these equations reduce to a single equation



$$\partial_t^2 \vec{\xi} + \nabla \phi + 2\vec{\Omega} \times \partial_t \vec{\xi} + \frac{\nabla P}{\rho} = 0, \tag{2}$$

where $\vec{\xi} = \vec{\xi}(t,\vec{a})$ is the displacement of a fluid particle from its original location at $\vec{a}$. The entropy, which for a given fluid

particle is conserved, is determined by its initial spatial distribution $s_0$:

$$s = s_0(\vec{a}).$$

The density $\rho$ in Eq. (2) follows from mass conservation and is given by the equation:

$$\rho(t,\vec{a}) = \rho_0(\vec{a}) \left| \frac{\partial(\vec{r})}{\partial(\vec{a})} \right|^{-1} = \rho_0(\vec{a}) \left| \frac{\partial(\vec{a}+\vec{\xi})}{\partial(\vec{a})} \right|^{-1} = \rho_0(\vec{a}) \left| 1 + \frac{\partial(\vec{\xi})}{\partial(\vec{a})} \right|^{-1}, \tag{3}$$

where $\vec{r} = \vec{a} + \vec{\xi}$ is the Cartesian coordinate of a fluid particle, $|\ |$ denotes the determinant of the corresponding matrix, and

$\rho_0(\vec{a})$ is the spatial distribution of density at $t = 0$. Equation (2) represents Newton's second law. Note that the gradients in

this equation are taken with respect to the Cartesian $\vec{r}$ rather than $\vec{a}$ coordinates.

The action $A$ in the general case is defined as follows:

$$A = \int_{t_0}^{t_1} L_*(q_i, \dot{q}_i) dt, \tag{4}$$

where $q_i$ is a set of arbitrary parameters characterizing the system state and $\dot{q}_i$ are their time derivatives. The index $i$ can be

either discrete or continuous. The least action principle

$$\delta A = 0 \tag{5}$$

results in the following (Euler-Lagrange) evolution equations:

$$\frac{d}{dt} \frac{\partial L_*}{\partial \dot{q}_i} - \frac{\partial L_*}{\partial q_i} = 0. \tag{6}$$

We now demonstrate that equation Eq. (2) follows from the LAP Eq. (5) for

$$L_*(\vec{\xi}, \partial_t \vec{\xi}) = \int L(\vec{\xi}, \partial_t \vec{\xi}, \vec{a}) d\vec{a}$$

with the Lagrangian density $L$ defined as follows:

$$L(\vec{\xi}, \partial_t \vec{\xi}, \vec{a}) = \rho_0(\vec{a}) \left\{ \frac{1}{2} (\partial_t \vec{\xi})^2 - E(\alpha, s_0(\vec{a})) - g(\vec{n}\vec{\xi}) - \vec{\Omega}(\partial_t \vec{\xi} \times \vec{\xi}) \right\}. \tag{7}$$

In Eq. (7) $E(\alpha, s)$ is the internal energy of the dry air per unit mass, $\vec{n}$ is a unit vector directed upwards along the gradient of

geopotential, and $g$ is its magnitude.

Let us calculate variation of the total energy with respect to $\vec{\xi}$ taking into account that according to the basic thermodynamic

identity $Tds = dE + Pd\alpha$ one has $P = -(\partial E / \partial \alpha)_s$:





$$\delta \int \rho_0(\vec{a}) E(\alpha, s_0(\vec{a})) d\vec{a} = \int \rho_0 \frac{\partial E}{\partial \alpha} \delta \alpha d\vec{a} = -\int \rho_0 P \delta \left( \frac{1}{\rho_0(\vec{a})} \left| \frac{\partial \vec{r}}{\partial \vec{a}} \right| \right) d\vec{a} =$$

$$= -\int P \sum_{i,j=1}^{3} C_{ij} \frac{\partial}{\partial a_j} \delta \xi_i d\vec{a} = -\int (\vec{N} \delta \vec{\xi}) P d\Sigma + \int \sum_{i,j=1}^{3} \frac{\partial P}{\partial a_j} C_{ij} \delta \xi_i d\vec{a}$$

Here $C_{ij}$ is the cofactor of the $(i, j)$-th entry of matrix $\partial \vec{r} / \partial \vec{a}$ (i.e. the determinant of this matrix with the $i$-th row and $j$

-th column deleted multiplied by $(-1)^{i+j}$). Integration by parts (i.e. use of the Gauss theorem) leads to appearance of the first

(surface) term with $\vec{N}$ being a unit outward normal to the boundary of the medium $\Sigma$. It is easy to make sure that

$\sum_j \partial C_{ij} / \partial a_j = 0$; for this reason, the derivatives of $C_{ij}$ in the equation above are absent. Considering that the matrix of

cofactors transposed is proportional to the inverse matrix one has:

$$C_{ij} = \left| \frac{\partial \vec{r}}{\partial \vec{a}} \right| \left( \left( \frac{\partial \vec{r}}{\partial \vec{a}} \right)^{-1} \right)_{ji} = \left| \frac{\partial \vec{r}}{\partial \vec{a}} \right| \frac{\partial a_j}{\partial r_i}$$

Taking into account that according to Eq. (3) $|\partial \vec{r} / \partial \vec{a}| = \rho_0 / \rho$ one finds:

$$\sum_{j=1}^{3} \frac{\partial P}{\partial a_j} C_{ij} = \frac{\rho_0}{\rho} \sum_{j=1}^{3} \frac{\partial P}{\partial a_j} \frac{\partial a_j}{\partial r_i} = \frac{\rho_0}{\rho} \frac{\partial P}{\partial r_i}$$

where now $P$ in the last equation above is considered is a function of $\vec{r}$ rather than $\vec{a}$. As a result, we obtain:

$$\delta \int \rho_0(\vec{a}) E(\alpha, s_0(\vec{a})) d\vec{a} = -\int (\vec{N} \delta \vec{\xi}) P d\Sigma + \int \frac{\rho_0}{\rho} \frac{\partial P}{\partial \vec{r}} \delta \vec{\xi} d\vec{a} \qquad (8)$$

Calculating now the variation of the action Eqs. (4), (7) and integrating terms proportional to $\partial_t \delta \vec{\xi}$ by parts with respect to time

yields:

$$\delta A = -\int dt d\vec{a} \rho_0(\vec{a}) \left( \partial_t^2 \vec{\xi} + g\vec{n} + 2\vec{\Omega} \times \partial_t \vec{\xi} + \frac{1}{\rho} \frac{\partial P}{\partial \vec{r}} \right) \delta \vec{\xi} + \int (\vec{N} \delta \vec{\xi}) P d\Sigma dt \qquad (9)$$

We can see that the requirement $\delta A = 0$ in Eq. (9) for internal points in fact coincides with Eq. (2). The internal energy of dry

air $E = C_V T$ in terms of the variables $(\alpha, s)$ is given by

$$E(\alpha, s) = \frac{p_{00} \alpha_{00}^{\gamma}}{\gamma - 1} \alpha^{1-\gamma} e^{s/C_V}, \qquad (10)$$

where $p_{00}, \alpha_{00}$ are reference values of pressure and specific volume and $\gamma = C_p / C_V$ is as usual the ratio of specific heats.

To derive a set of numerically accessible dynamical equations that adequately describe the evolution of the continuous atmosphere

requires one additional standard step. One difficulty is that the LHS of Eq. (6) in the discrete case will generally contain a mix of

second derivatives of the discrete coordinates $\ddot{q}_i$ with different indices, and not being resolved with respect to $\ddot{q}_i$ the resulting



equations are ill-suited for numerical solution. This issue can be resolved by representing Eq. (6) in the Hamiltonian form. Namely,

we introduce the momenta $p_i$ according to the relation

$$p_i = \frac{\partial L_*}{\partial \dot{q}_i} \tag{11}$$

and express from these equations $\dot{q}_i$ as functions of $q, p$. From this point forward, the variable $p$ will stand for conjugated

momentum. We introduce instead of the Lagrangian $L_*(q, \dot{q})$ the Hamiltonian $H_*(q, p)$ according to the equation

$$H_*(q, p) = \sum_i \dot{q}_i p_i - L_*(q, \dot{q}), \tag{12}$$

where all $\dot{q}_i$ have been expressed as functions of $q, p$ as noted above. It is not difficult to show by using Eqs. (11) and (12) that

the following (Hamiltonian) equations hold:

$$\dot{q}_i = \frac{\partial H_*}{\partial p_i}, \quad \dot{p}_i = -\frac{\partial H_*}{\partial q_i}. \tag{13}$$

These expressions follow by applying the LAP Eq. (5) to

$$A = \int_{t_0}^{t_1} \left\{ \sum_i \dot{q}_i p_i - H_*(q, p) \right\} dt \tag{14}$$

and varying independently the variables $q_i$ and $p_i$. In what follows we will only be using LAP in this form.

Substituting Eq. (7) into Eq. (11) we obtain:

$$\vec{p}(t, \vec{a}) = \frac{\partial L\left(\vec{\xi}, \partial_t \vec{\xi}, \vec{a}\right)}{\partial \left(\partial_t \vec{\xi}\right)} = \rho_0(\vec{a}) \left(\partial_t \vec{\xi} + \vec{\Omega} \times \vec{\xi}\right). \tag{15}$$

Expressing $\partial_t \vec{\xi}$ in terms of $\vec{p}, \vec{\xi}$ and substituting the result into Eq. (12) after simple transformations for the density of the

Hamiltonian yields


$$H\left(\vec{\xi}, \vec{p}; \vec{a}\right) = \frac{\left(\vec{p} - \rho_0(\vec{a})\vec{\Omega} \times \vec{\xi}\right)^2}{2\rho_0(\vec{a})} + \rho_0(\vec{a}) E\left(\alpha, s_0(\vec{a})\right) + g \rho_0(\vec{a}) \left(\vec{n}\vec{\xi}\right) \tag{16}$$

The Hamiltonian equations (13) read:

$$\partial_t \vec{\xi} = \frac{\delta \tilde{H}}{\delta \vec{p}} = \frac{\vec{p} - \rho_0(\vec{a})\vec{\Omega} \times \vec{\xi}}{\rho_0(\vec{a})}$$

$$\partial_t \vec{p} = -\frac{\delta \tilde{H}}{\delta \vec{\xi}} = \left(\vec{p} - \rho_0(\vec{a})\vec{\Omega} \times \vec{\xi}\right) \times \vec{\Omega} - g \rho_0(\vec{a})\vec{n} - \frac{\rho_0(\vec{a})}{\rho} \frac{\partial P}{\partial \vec{r}}$$

By differentiating the first of these equations with respect to time and excluding from the result $\partial_t \vec{p}$ using the second equation

we make sure that the result coincides in fact with Eq. (2).

Finally, the expression Eq. (14) for the action for the continuous atmosphere reduces to



$$A = \int_{t_0}^{t_1} dt \int d\vec{a} \left\{ \vec{p}\partial_t\vec{\xi} - \frac{\left(\vec{p} - \rho_0\left[\vec{\Omega},\vec{\xi}\right]\right)^2}{2\rho_0(\vec{a})} - \rho_0(\vec{a})E(\alpha,s_0(\vec{a})) - g\rho_0(\vec{a})(\vec{n}\vec{\xi}) \right\}. \tag{17}$$

## 3 The action for a discrete atmospheric model

We now need to select a set of discrete parameters that represent a continuous atmosphere and a mode of interpolating these
parameters to an arbitrary spatial point. The model that will be used here is as follows. We split the region of interest into a set of unstructured connected tetrahedra that share vertices, edges, and faces. We select values for the displacements $\vec{\xi}$ and momenta $\vec{p}$, as well as the density $\rho_0$ and specific entropy $s_0$, at the vertices, which define a set of discrete parameters that represent the continuous atmosphere. To calculate corresponding parameters at an arbitrary point within each tetrahedron we will be using linear interpolation. We note that four values of a parameter at the four vertices of a tetrahedron completely determine its linear

interpolation within the tetrahedron. The pricewise linear interpolation ensures a globally continuous representation of the corresponding fields throughout the region. Derivatives along the faces of tetrahedra are also continuous, however, derivatives across the faces experience jumps, which for a sufficiently dense set of tetrahedra are negligible. Let us note also that such interpolation ensures calculation of the action to the accuracy of square of the ratio of a linear size of tetrahedra to a spatial scale of variations of atmospheric parameters.

We can now approximate the action $A$ by $\tilde{A}$ where

$$A \approx \tilde{A} = \sum_T A_T, \tag{18}$$

where the index $T$ stands for an individual tetrahedron and the summation aggregates the contribution from all tetrahedrons. The action $A_T$ is calculated according to Eq. (17), where the integral is over the volume occupied by the $T$-th tetrahedron.

Let us consider the first term $\vec{p}\partial_t\vec{\xi}$ in Eq. (17). The following relation can be shown to hold true for two linear functions of
coordinates $u(\vec{a})$ and $v(\vec{a})$ defined within a tetrahedron by linear approximations based on their values $u_i$, $v_i$, $i = 1, 2, 3, 4$ at the vertices of the tetrahedron:

$$\int_T u(\vec{a})v(\vec{a})d\vec{a} = \frac{|V_T|}{4}\left(\sum_{i=1}^{4}u_iv_i - \frac{1}{2}\sum_{i,j=1}^{4}(u_i - u_j)(v_i - v_j)\right). \tag{19}$$

Here $|V_T|$ is the tetrahedron's volume. For smooth fields, the differences in the second term in the RHS of Eq. (19) will be proportional to the ratio of the linear size of the tetrahedron to the characteristic scale of the parameters variation, and the magnitude
of the entire second term will be proportional to the square of this ratio and can be generally neglected when compared to the first term in Eq. (19). We then find:

$$\int_T \vec{p}\partial_t\vec{\xi}d\vec{a} \approx \frac{|V_T|}{4}\sum_{i=1}^{4}\vec{p}_i\partial_t\vec{\xi}_i. \tag{20}$$



We note, however, that the neglected term can be retained by replacing (to a first approximation) the differences $\partial_t \vec{\xi}_i - \partial_t \vec{\xi}_j$ in

(19) by $\partial H / \partial \vec{p}_i - \partial H / \partial \vec{p}_j$. This will result in the following modification to the Hamiltonian density in Eq. (16) within

the corresponding tetrahedron:

$$H\left(\vec{\xi},\vec{p};\vec{a}\right) \to H\left(\vec{\xi},\vec{p};\vec{a}\right) - \frac{1}{8}\sum_{i,j=1}^{4}\left(\vec{p}_i - \vec{p}_j\right)\left(\frac{\partial H}{\partial \vec{p}_i} - \frac{\partial H}{\partial \vec{p}_i}\right).$$

The transition from a continuous to discrete description of the mechanical system based on the approximation of action, which is pursued here, differs from the more customary approximation of a continuous Hamiltonian by a discrete analog (Salmon, 1983). Using Eq. (20) we can represent the action as

$$\tilde{A} = \int_{t_0}^{t_1} dt \sum_T |V_T| \left(\frac{1}{4}\sum_{n(T)}\vec{p}_{n(T)}\partial_t\vec{\xi}_{n(T)} - \tilde{H}_T\left(\vec{\xi}_{n(T)},\vec{p}_{n(T)}\right)\right), \tag{21}$$

where

$$\tilde{H}_T\left(\vec{\xi}_{n(T)},\vec{p}_{n(T)}\right) = \frac{1}{|V_T|}\int_T H\left(\vec{\xi},\vec{p};\vec{a}\right)d\vec{a} \tag{22}$$

is the Hamiltonian in Eq. (16) integrated over the $T$-th tetrahedron and normalized by its volume. $\tilde{H}_T$ is a function of shifts and

momenta at the vertices of the tetrahedron $\vec{\xi}_{n(T)}$ and $\vec{p}_{n(T)}$, where $n(T)$ stands for indices of the set of four vertices of the $T$

-th tetrahedron. We now replace in Eq. (21) the summation over tetrahedra with a summation over vertices and represent it in the following form:

$$\tilde{A} = \int_{t_0}^{t_1} dt \sum_k \left(v_k \vec{p}_k \partial_t \vec{\xi}_k - \sum_{\tau \subset T(k)} |V_\tau| \tilde{H}\left(\vec{\xi}_{n(\tau)},\vec{p}_{n(\tau)}\right)\right). \tag{23}$$

Here $T(k)$ represents indices of the tetrahedra that contain the vertex with index $k$, and the summation in the parenthesis in

Eq. (23) aggregates the contribution over all such tetrahedra. The parameter $v_k$ is a quarter of the sum of their volumes:

$$v_k = \frac{1}{4}\sum_{\tau \subset T(k)}|V_\tau|. \tag{24}$$

Equation (23) describes an approximation of the action for a continuous atmosphere in terms of a finite set of discrete parameters.

We can now write the equation for the evolution of these parameters by minimizing $\tilde{A}$ in Eq. (23) as:

$$\partial_t \vec{\xi}_k = \frac{1}{v_k}\sum_{\tau \subset T(k)}|V_\tau|\frac{\partial}{\partial \vec{p}_k}\tilde{H}_\tau\left(\vec{\xi}_{n(\tau)},\vec{p}_{n(\tau)}\right),$$

$$\partial_t \vec{p}_k = -\frac{1}{v_k}\sum_{\tau \subset T(k)}|V_\tau|\frac{\partial}{\partial \vec{\xi}_k}\tilde{H}_\tau\left(\vec{\xi}_{n(\tau)},\vec{p}_{n(\tau)}\right). \tag{25}$$

Note that the action in Eq. (23) can be cast precisely into the form Eq. (14) by rescaling coordinates and momenta thus making the

corresponding system explicitly Hamiltonian although this step is not needed here. Since the approximate action is invariant with





respect to shifts in time the approximate energy is conserved. Similarly, if the exact action is invariant with respect to geometrical transformations like shifts in space or rotations, the approximate action will inherit this property along with corresponding conservation laws; conservation will be exact; however, the conserved values will be calculated approximately.

Equations (25) can be solved numerically using a suitable integration scheme (e.g., Runge-Kutta). There are also time-integration
schemes that conserve energy exactly (namely, symplectic (Eldred et al, 2019) and variational (Mardsen and West, 2001; Lew et al, 2003)).

To be able to repeat the time step we have to reassign values of momenta $\vec{p}_k$ at corresponding vertices whose locations $\vec{a}_k$ are fixed in space. We emphasize that $\vec{\xi}_k$ and $\vec{p}_k$ in Eq. (25) are Lagrangian coordinates and $\vec{p}_k(\Delta t)$ is the momentum of a fluid particle which at $t = \Delta t$ is located at a point $\vec{r}_k = \vec{a}_k + \vec{\xi}_k(\Delta t)$ and not at $\vec{a}_k$. For this reason, we have to first determine
which particle moved to $\vec{a}_k$ at $t = \Delta t$, and then calculate its momentum using the forward operator (i.e., linear interpolation). This value will be an initial condition for $\vec{p}_k$ in Eq. (25) for the next time step. The initial values of entropy and passive tracers will also correspond to the fluid particle that moved to $\vec{a}_k$. These should be reassigned as well. The initial values of shifts are always $\vec{\xi}_k = 0$. By making these reassignments we are essentially returning to the Eulerian description of fluid motion.

To determine the initial location of a fluid particle that arrived at a point $\vec{R}$ at a time $t = \Delta t$, we proceed as follows. The
position of an arbitrary point within a tetrahedron can be expressed as follows:

$$\vec{r} = \vec{a}_4 + (\vec{a}_1 - \vec{a}_4)\tau_1 + (\vec{a}_2 - \vec{a}_1)\tau_2 + (\vec{a}_3 - \vec{a}_2)\tau_3, \tag{26}$$

where $\vec{a}_i$ are the Cartesian coordinates of the $i$-th vertex of the tetrahedron, and the fourth vertex was arbitrarily selected in Eq. (26) as a base point. For the point $\vec{r}$ to be within the tetrahedron, the scalar dimensionless parameters $\tau_i$ (which have nothing to do with $\tau$ in Eqs. (24), (25)) should satisfy

$$1 > \tau_1 > \tau_2 > \tau_3 > 0. \tag{27}$$

Due to the assumption of a linear dependence of the shifts $\vec{\xi}$ on the Cartesian coordinates within a tetrahedron, the fluid particle located at $\vec{r}$ at $t = 0$ will at the end of a time step at $t = \Delta t$ experience a shift given by an expression similar to Eq. (26), that is:

$$\vec{\xi} = \vec{\xi}_4 + (\vec{\xi}_1 - \vec{\xi}_4)\tau_1 + (\vec{\xi}_2 - \vec{\xi}_1)\tau_2 + (\vec{\xi}_3 - \vec{\xi}_2)\tau_3, \tag{28}$$

where $\vec{\xi}_i$, $i = 1,..,4$ are shifts of fluid particles located at corresponding vertices at $t = 0$; these shifts are known as a result of numerical integration of Eq. (25). The requirement of the fluid particle to translate from a point $\vec{r}$ to a point $\vec{R}$ is given by:

$$\vec{r} + \vec{\xi} = \vec{R}.$$

Substituting into this equation Eqs. (26) and (28) we obtain the following vector linear equation with respect to the parameters $\tau_i$ :



$$\sum_{i=1}^{3} \left( \vec{a}_i - \vec{a}_4 + \vec{\xi}_i - \vec{\xi}_4 \right) \tau_i = \vec{R} - \vec{a}_4 - \vec{\xi}_4. \tag{29}$$

The fluid particle can in principle arrive at the $k$-th vertex from any tetrahedron from the set $T(k)$, however, the condition in Eq. (27) selects the appropriate tetrahedron. The existence and uniqueness of such solution follows from the fact that the shifts $\vec{\xi}_i$ are expected to be much smaller than the sizes of the tetrahedra so that the topological structure of the initial and shifted points doesn't change.

The value of any parameter within a tetrahedron at the end of a time step is given by an expression quite similar to Eq. (28). For example, the momentum at a time $t = \Delta t$ at a vertex with Cartesian coordinate $\vec{R}$ is given by:

$$\vec{p} = \vec{p}_4 + \left( \vec{p}_1 - \vec{p}_4 \right) \tau_1 + \left( \vec{p}_2 - \vec{p}_1 \right) \tau_2 + \left( \vec{p}_3 - \vec{p}_2 \right) \tau_3 \tag{30}$$

where $\vec{p}_i = \vec{p}_i \left( \Delta t \right)$ are the momenta which are also known as a result of numerical integration of Eq. (25). The parameters $\tau_i$ in Eq. (30) follow from solution of Eq. (29). The resulting $\vec{p}$ in Eq. (30) is the momentum which has to be reassigned to
corresponding vertex as an initial condition for the next time step. The specific entropy at the vertices should be also redefined according to Eq. (30), where now the values of $s_0$ at $t = 0$ should be substituted for $\vec{p}_i$ (since the entropy of the fluid particle is conserved). All other passive tracers should be similarly redefined. The value of the density $\rho_0$ at the $k$-th vertex should be redefined using Eq. (3) with respect to the same tetrahedron and the same fluid particles that were used for recalculating the momentum and entropy (i.e., with the same $\tau_i$ following from Eq. (29)).

We consider now the calculation of the Hamiltonian $\tilde{H}$, which forms the basis of the numerical model. We note that Eq. (25) describes the evolution of the model parameters and includes the derivatives of $\tilde{H}$. For simplicity, we assign here to the vertices of a tetrahedron the indices $n = 1, 2, 3, 4$. We now recast the integration over $\vec{a}$ in Eq. (2) into an integration over the dimensionless parameters $\tau_{1,2,3}$ according to Eq. (26). The functions $\vec{\xi}(\vec{a})$, $\vec{p}(\vec{a})$, $\rho_0(\vec{a})$, $s_0(\vec{a})$ are calculated according to the selected forward operator, i.e. by linear approximation quite similar to Eqs. (28) and (30):

$$f(\tau_i) = f_4 + \left( f_1 - f_4 \right) \tau_1 + \left( f_2 - f_4 \right) \tau_2 + \left( f_3 - f_4 \right) \tau_3, \tag{31}$$

where $f$ is any parameter (vector or scalar) with $f_n$ being a value of $f$ at the $n$-th vertex.

According to Eq. (25) we have:

$$\tilde{H}_T \left( \vec{\xi}_n, \vec{p}_n \right) = 6 \int_0^1 d\tau_1 \int_0^{\tau_1} d\tau_2 \int_0^{\tau_2} d\tau_3 H \left( \vec{\xi}(\tau_n), \vec{p}(\tau_n); \tau_n \right), \tag{32}$$

where $H$ is given by Eq. (16) and $\vec{\xi}(\tau_n)$, $\vec{p}(\tau_n)$ by Eq. (31). After the LHS of Eq. (32) becomes a function of the values
of the displacements $\vec{\xi}$ and momenta $\vec{p}$ at the vertices of the tetrahedron. Note that since $\vec{\xi}$ is a linear function of coordinates $\vec{a}$, the Jacobian in Eq. (3) is a constant independent of $\vec{a}$ and we can write



$$\alpha(\tau_n) = \frac{\Delta_0(\vec{\xi}_n)}{\rho_0(\tau_n)},$$

where $\Delta_0(\vec{\xi}_n)$ is the value of the Jacobian. The internal energy in Eq. (16) can then be expressed as:

$$\rho_0 E(\alpha, s_0) = \frac{p_{00}\alpha_{00}^{\gamma}}{\gamma - 1}\left(\Delta_0(\vec{\xi})\right)^{1-\gamma}\left(\rho_0(\tau_n)\right)^{\gamma}\exp\left(\frac{s_0(\tau_n)}{C_V}\right). \tag{33}$$

To calculate the integral in Eq. (32) related to the internal energy term we note that variations of density within a tetrahedron are generally small and set

$$\left(\rho_0(\tau_n)\right)^{\gamma} = \rho_4^{\gamma}\left(1 + \left(\sum_{n=1}^{3}\beta_n\tau_n\right)\right)^{\gamma} =$$

$$= \rho_4^{\gamma}\left(1 + \gamma\left(\sum_{n=1}^{3}\beta_n\tau_n\right) + \frac{\gamma(\gamma-1)}{2}\left(\sum_{n=1}^{3}\beta_n\tau_n\right)^2 + ...\right), \tag{34}$$

where the factors

$$\beta_n = \frac{\rho_n - \rho_4}{\rho_4} \tag{35}$$

are also small, so that one only needs to retain the first few expansion terms in Eq. (34). One can similarly expand into a power series the exponential term in Eq. (33). In this case calculation of the integral in Eq. (32) with respect to the internal energy term also reduces to an integration of polynomials of $\tau_{1,2,3}$. Alternatively, one can leave the exponential term in Eq. (33) as is since the exponent is a linear function of $\tau_{1,2,3}$ and integrals of products of polynomials and exponentials can be calculated analytically nearly as easily as integrals of polynomials. In the first term in the RHS of Eq. (16) (the kinetic energy) we also expand

$1/\rho_0(\tau_i)$ according to Eq. (34), where now in place of $\gamma$ we substitute $-1$. Then the integrations over $\tau_{1,2,3}$ in this term also reduce to an integration of polynomials and is done analytically.

We consider now boundary conditions that follow from the LAP. They are due to the last (surface) term in Eq. (9) and require that at the boundaries either the pressure or $\left(\vec{N}, \delta\vec{\xi}\right)$ be zero. Thus, without restrictions on the variations of shifts of the boundary points, the pressure at the boundary should be zero. Such a boundary condition can be applied reasonably to the top of the

atmosphere (the issue of absorbing layer is beyond the scope of this paper) but is unacceptable elsewhere. For such boundary points we will assume that both the shifts $\vec{\xi}_k$ and momenta $\vec{p}_k$ are prescribed functions of time so that $\delta\vec{\xi}_k = 0$. Equations for the corresponding $k$ are excluded from the set Eq. (25). Note that $\vec{p}$ and $\vec{\xi}$ are related through Eq. (15), and it is sufficient to prescribe a time dependence to only one or the other. In particular, at the bottom of the atmosphere one can use the no-slip boundary condition $\vec{\xi} = 0$. To introduce different boundary conditions, one would need to add corresponding terms to the

expression for the action.



Our numerical approach proceeds as follows. We introduce an array of tendencies (i.e., the time derivatives $\partial_t \vec{\xi}_k, \partial_t \vec{p}_k$ ) with the total number of columns equal to the total number of vertices - one column per vertex. To calculate the RHS of the evolution equations in Eq. (25), we initiate a loop not over vertices but over tetrahedra. After calculating the derivatives $\partial \tilde{H}_\tau / \partial \vec{p}_k$ and $\partial \tilde{H}_\tau / \partial \vec{\xi}_k$ for all four vertices of a particular tetrahedron (which results in eight vectors with a total of 24 scalar parameters),

we add/subtract the result to/from the tendencies being accumulated in the corresponding columns of the tendencies array. After all the tetrahedrons are accounted for, the RHS of Eq. (25) will appear automatically. To execute this procedure, we also introduce an integer array with the same number of columns, with each column containing four indices of vertices belonging to a corresponding tetrahedron.

The algorithm under consideration easily allows us to modify the set of tetrahedra during simulations. If after a time step one finds
that at a particular area spatial resolution should be increased, the corresponding tetrahedra are split into two. Each split tetrahedron is flagged, and two new tetrahedra are added. This operation can be repeated as many times as necessary. Similarly, previously split tetrahedra can be recombined into larger ones by reversing the process. The evolution is then calculated with respect to the modified set of tetrahedra.

**4 A numerical example**

The algorithm under consideration was tested for a 2D case for simplicity. The Hamiltonian structure can be introduced into the hydrodynamics in many different ways (Salmon, 1988), and the formulation used in this section is significantly different from the one used in Secs. 2-3. It is important, however, that the essence of the approach which is advocated in this paper is still the same:
approximation of the action using linear interpolation of canonical variables within corresponding simplexes. In the 2D case tetrahedral are replaced by triangles and volumes by corresponding areas.

For historical reasons the following formulation was used in this section:

$$H = \int dV \rho \left[ \frac{1}{2}\vec{v}^2 + gz + E(\rho,s) + \Psi(s) \right], \tag{36}$$

where

$$\vec{v} = \nabla\varphi - \frac{\lambda}{\rho}\nabla s \tag{37}$$

is a velocity, $(\rho,s)$ are the density and specific entropy, respectively, which play the role of coordinates, and $(\varphi,\lambda)$ are conjugated momenta, which do not have a direct physical interpretation. A gauge function $\Psi(s)$ is chosen to ensure hydrostatic equilibrium for the corresponding model of the atmosphere. The Hamiltonian Eq. (36) was suggested in (Goncharov et al, 1979), where the function $\Psi(s)$ corresponded to the isothermal atmosphere. Here this function was chosen for the more realistic case
of an atmosphere with a constant temperature gradient:



$$T(z) = T_{00}\left(1 - \frac{z}{H}\right).$$

(38)

In this case one sets

$$\Psi(s) = -gH + \left(gH - C_p T_{00}\right)\exp\left(-\frac{s}{\beta_0 C_V}\right),$$

(39)

where

$$\beta_0 = \frac{gH}{C_v T_{00}} - \gamma.$$


The governing equations for the continuous case are as follows:

$$\partial_t \rho = \frac{\delta H}{\delta \varphi} = -\nabla(\rho \vec{v}), \quad \partial_t \varphi = -\frac{\delta H}{\delta \rho} = -\vec{v}\nabla\varphi + \frac{\vec{v}^2}{2} - gz - \left(\frac{\partial \rho E}{\partial \rho}\right)_s - \Psi(s),$$

$$\partial_t s = \frac{\delta H}{\delta \lambda} = -\vec{v}\nabla s, \qquad \partial_t \lambda = -\frac{\delta H}{\delta s} = -\nabla(\lambda \vec{v}) - \rho T - \rho \Psi'(s).$$

(40)

Similar to the approach presented in the previous sections, we split the atmosphere into a set of triangles and linearly interpolate

all four prognostic variables $(\rho, s, \varphi, \lambda)$ within the triangles. Approximation Eq. (20) for the action was used, where the factor

of 1/4 was replaced by 1/3 and the volume by the area. The resulting discrete evolution equations are quite similar to Eq. (25) and

are as follows:

$$\partial_t \rho_k = \frac{1}{\sigma_k} \sum_{\chi \subset T(k)} \left|\Sigma_\chi\right| \frac{\partial \tilde{H}_\chi}{\partial \varphi_k}, \quad \partial_t \varphi_k = -\frac{1}{\sigma_k} \sum_{\chi \subset T(k)} \left|\Sigma_\chi\right| \frac{\partial \tilde{H}_\chi}{\partial \rho_k},$$

$$\partial_t s_k = \frac{1}{\sigma_k} \sum_{\chi \subset T(k)} \left|\Sigma_\chi\right| \frac{\partial \tilde{H}_\chi}{\partial \lambda_k}, \quad \partial_t \lambda_k = -\frac{1}{\sigma_k} \sum_{\chi \subset T(k)} \left|\Sigma_\chi\right| \frac{\partial \tilde{H}_\chi}{\partial s_k},$$

(41)

where now

$$\tilde{H}_\chi(\rho_n, s_n, \varphi_n, \lambda_n) = 2\int_0^1 d\tau_1 \int_0^{\tau_1} d\tau_2 H\left(\rho(\tau_n), s(\tau_n), \varphi(\tau_n), \lambda(\tau_n); \tau_n\right).$$

(42)

Here $H$ is given by Eq. (36) and $\chi, T(k)$ corresponds to triangles instead of tetrahedra with

$$\sigma_k = \frac{1}{3} \sum_{\chi \subset T(k)} \left|\Sigma_\chi\right|,$$

where $\left|\Sigma_\chi\right|$ represents the area of the $\chi$-th triangle.

We note that the Hamiltonian in Eq. (36) can be applied to the 3D case as well. For this case, however, we have only two free

parameters $\varphi$ and $\lambda$ to represent the velocity field in Eq. (37), and this is generally insufficient. This problem can be resolved

by introducing two additional parameters into the representation for the velocity in Eq. (37):




$$\vec{v} = \nabla \varphi - \frac{\lambda}{\rho} \nabla s + \zeta \nabla \mu.$$

The form of Hamiltonian in Eq. (36) and the procedure for calculating the action don't change, and another pair of equations needs to be added to Eq. (40) that correspond to the conservation of $\zeta$ and $\mu$ along the trajectories of fluid particles. Still, the Hamiltonian in Eq. (16) seems to be a better option.

For our simulation, a standard atmosphere was chosen with an effective height $H = 4.5 \cdot 10^4 \, m$ and surface temperature $T_{00} = 288 \, K$ (see Eq. (38), in this case $\beta_0 \approx 0.75$). A section of the atmosphere with a total length of $500 \, km$ and height of $3 \, km$ was split initially into 25 layers in the horizontal and 15 layers in the vertical, which resulted in 750 triangles. To simulate a convective event, we added to the Hamiltonian density in Eq. (36) a term $q\lambda$ so that the equation for entropy in Eq. (40) (the first equation in the second line) included in the RHS an additional term equal to $q$ (a local heat source). We assumed

that the heat source had a Gaussian shape, both in space and time, with characteristic scales of $100 \, km$ in the horizontal, $500 \, m$ in vertical, and $200 \, s$ in time. The source was centred at a height of $1 \, km$. The maximal intensity of the heat source was chosen to be $0.6 \, W / kg$, which corresponds to a rain rate of $1 \, mm / h$. To simulate the effects of topography, a hill of $500 \, m$ height and stretching for $80 \, km$ in the horizontal (also of Gaussian shape) centred at $125 \, km$ was included. Our simulation lasted for $600 \, s$, and the fourth-order Runge-Kutta method was used for integration. To check the functionality of

the algorithm with respect to the splitting/recombination of the triangles, the code was forced to split them if the difference in the intensity of the heat source at the vertices exceeded a certain threshold, and recombine them if the difference was less than another threshold. The time step used $\Delta t = 0.25 \, s$ was rather small due to the fact that fast-propagating sound waves are fully accounted of. It was found that that mass was conserved to a machine precision; conservation of energy wasn't checked because the numerical example included heating of the atmosphere.

The isolines of entropy (i.e., potential temperature) are plotted in a series of images as solid lines in Fig. 1 for five snapshots in time. The triangles (a bit skewed due to presence of the hill) are shown by thin lines. The split triangles are emphasized. One can see the development of convection and appearance of the irregular structure within the convective cell that is due most likely to convective instability.

**5 Conclusions**


The dynamical core for simulation of atmospheric motion presented in this paper can be considered to be a finite-element method combined with the least-action principle. It includes features of both a Eulerian and Lagrangian description of fluid motion. The discrete set of parameters representing the atmosphere at the beginning and at the end of each time step correspond to the same set of spatial points with the associated fluid particles located at these points (i.e., a Eulerian description). The evolution within a time

step, however, makes use of the Lagrangian description of fluid motion. The advantages of this approach include: (1) For a given set of grid points and a given forward operator the algorithm ensures through a minimization of action a maximal closeness (in a broad sense) of the evolution of the discrete system to the motion of the continuous atmosphere (i.e., a dynamically-optimal algorithm); (2) The grid of selected discrete points can be irregular allowing for a variable resolution in space; (3) The spatial



resolution can be adjusted locally while executing calculations; (4) The use of a set of tetrahedra as finite elements in the algorithm
ensures a better (piecewise linear rather than staircase) representation of topography; (5) The algorithm automatically calculates
the evolution of passive tracers by following the trajectories of the fluid particles, which ensures that all *a priori* required passive
tracer properties are satisfied.

For demonstration purposes the algorithm presented here considers only the very simple case of a dry atmosphere and the evolution
of an arbitrary number of passive tracers. To consider a real atmosphere one has to add to the RHS of Eq. (25) heat sources and
turbulent stresses. To include heating/cooling of the atmosphere, passive tracers such as water vapor and hydrometeors need to be
added to the development. Their influence can be accounted for by considering the variation of entropy of the fluid particles at
each time step by including heat exchange processes at the grid vertices, or, more accurately if needed, with account of the motion
of the fluid particles. At this stage one would need information regarding brightness temperatures from corresponding radiative
transfer calculations. The effects of turbulence can also be included by adding Reynolds stresses $\tau_{ij}$ to the development through
an additional term

$$\sum_{i,j} \tau_{ij} \partial \xi_i / \partial a_j.$$

in action Eq. (21) and integration over the tetrahedra, which would provide the corresponding contributions to the evolution of
momenta in Eq. (25).

## 6 Competing interests

The contact author has declared that none of the authors has any competing interests.

## 7 Acknowledgements

The author expresses his gratitude to the NOAA Physical Sciences Laboratory for support of this work. The author is also very
much indebted to Dr. R.J. Lataitis, who carefully read the manuscript and made many valuable suggestions.

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

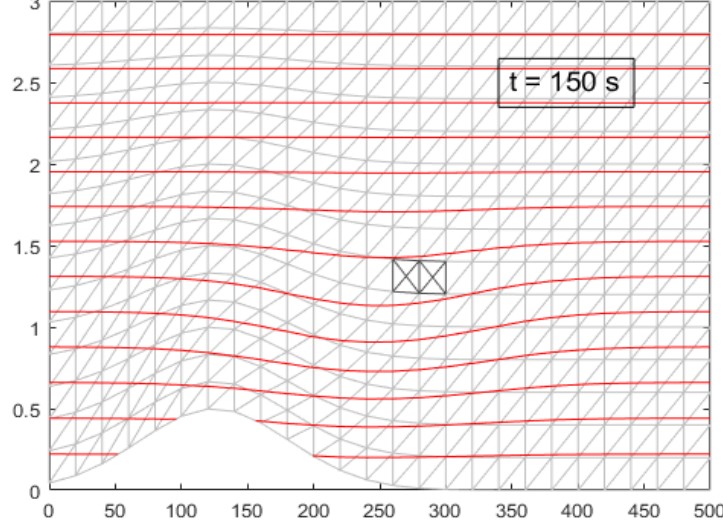



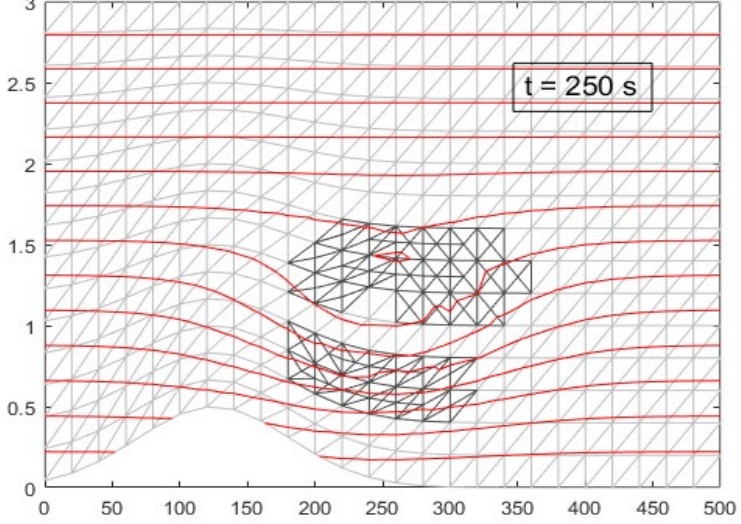

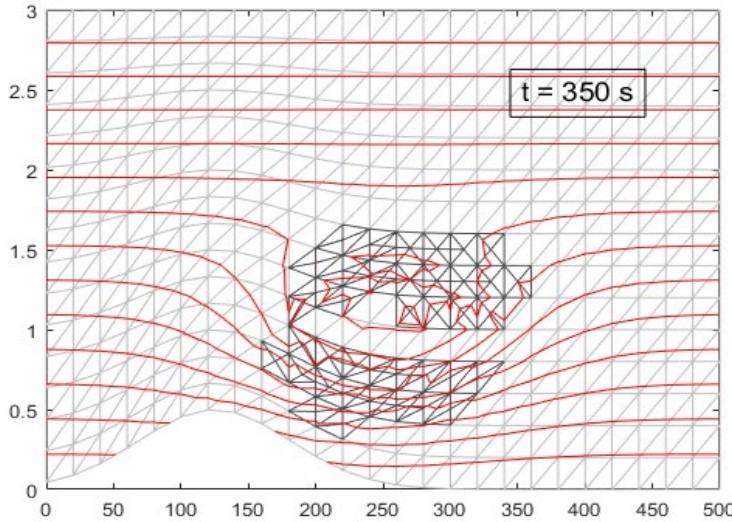



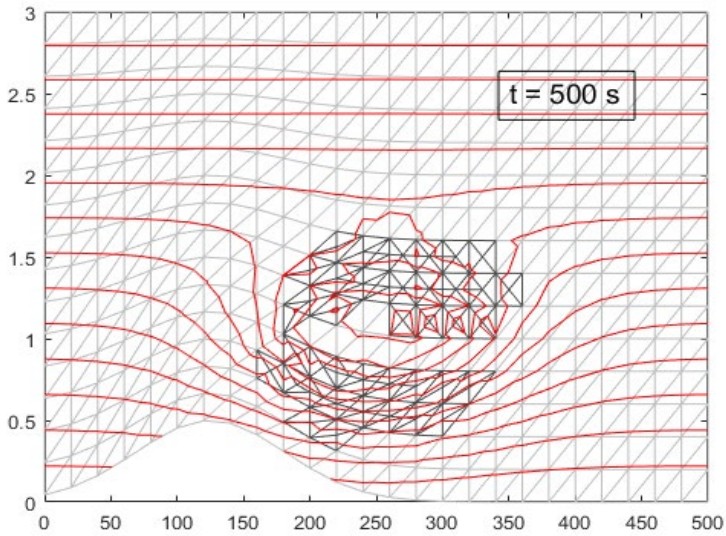


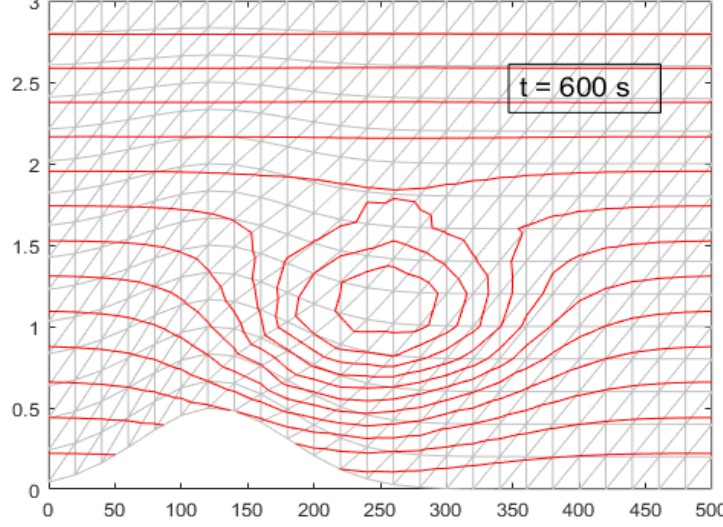

**Figure 1: Snapshots of the isolines of the entropy (potential temperature) field at selected times for the numerical example described in Section 4 (distances along the axes are in km).**