# Peer review of "Dynamically-optimal models of atmospheric motion"

_EGUsphere, 2024_

## Referee Comment (RC2)

This is a review of the mauscript entitled "Dynamically-optimal models of atmospheric motion" by A. G. Voronovich.

The paper presents a new treatment for the discretization of the equations of atmospheric dynamics based on the principle of least action, in which the action is itself discretized. The scheme is a hybrid of Lagrangian-Eulerian dynamics that enables exact numerical maintenance of the conservation laws admitted by the symmetries of Lagrangian via Noether's Theorem. The adoption of the Eulerian grid enables a discretization in terms of tetrahedral (or, in 2d, triangular) finite elements that provide a good treatment of topography. The description of the method in three dimensions is offered, which straight-forwardly covers the validation which is conducted in two dimensions. Finally, the implementation and the 2d example shows the ease with which grid refinement may be accommodated.

The paper is reasonably well organized, and well-written. The technique and its implementation are interesting, and the paper provides a much less ad-hoc representation of the way in which the discretized system can reflect the mimetic properties of the continuum systems–at least in terms of their conservation laws– than more fashionable (and usually Eulerian) treatments that *start* with the conservation laws, rather than with the underlying action principle. For these reasons, this paper should be published, as it makes an important contribution to the literature on the numerics of computational atmospheric dynamics that will have broader appeal to other computational fields.

That said, there are numerous items that should be addressed. Below, these items are split into "technical" and "editorial" considerations. Particularly regarding the former, I hope the author addresses these in the spirit in which they are offered: to help clarify and perhaps generalize the presentation. The editorial comments should largely be accepted so as to correct the grammar. I look forward to seeing this manuscript in its final state, and if a second reading is required, I am happy to provide one.

**1 Technical considerations**

1. Eq. (1): your transport term is missing a dot product; should be $\vec{v} \cdot \nabla \vec{v}$. Same for entropy transport. Also, for the conservation of mass, second term should read $\nabla \cdot (\rho \vec{v})$. This is a common problem throughout. I'll try to point these out, but please double-check this, as I may miss some. Even though it doesn't cause much confusion, it's an abuse of notation at a minimum, and should be corrected.

2. Eq (7): Rotation term must be a scalar!

3. Equation above line 95: Aren't you missing a $\sum C_{ij}$ term in the surface term? Also, the the integrand $\vec{N}\delta\vec{\xi}$ should be an inner product, but depending on whether you add the $\sum C_{ij}$, and use indices, you may not want a $\cdot$.

4. Eq (9): Again, integrands aren't manifestly scalar because you're missing inner product symbols.

5. Eq (9): I'm being pedantic here, but the integration by parts leads, I think, to a term:

$$\int d\vec{a} \int_{t_0}^{t_1} dt \partial_t [\delta\vec{\xi} \cdot (\partial_t \vec{\xi} + \vec{\xi} \times \vec{\Omega})],$$

so the principle of least action implies that either (a) $\delta\vec{\xi}$ must be orthogonal to $D_t\vec{\xi} \equiv \partial_t\vec{\xi} + \vec{\xi} \times \vec{\Omega}$ for any finite time interval, hence for all time, or that (b) $D_t\vec{\xi}|_{t_0} = D_t\vec{\xi}|_{t_1}$, assuming $D_t\vec{\xi} \neq 0$ at all times. Is this just ignored, or are there potential dynamical implications?

6. Eq (7), Eqs (25): It's often the case that we want to evolve the equations in a background hydrostatic state. Would you modify the Lagrangian to accommodate this by including new canonical variables (the density and pressure/energy fluctuation), or would you modify the Eqs (25) in a sort of ad-hoc way to accomplish this?

7. Eq (16): Shouldn't $H$ on the LHS be $\tilde{H}$, or else the $\tilde{H}$ in the two subsequent equations be just $H$? Also, there's an inner product symbol missing in the gravity term in Eq (16).

8. $\partial_t \vec{p}$ equation just below Eq (16): When computing the pressure gradient term, I find that:

$$-\rho_0 \frac{\partial E}{\partial \xi_i} = -\rho_0 \frac{\partial E}{\partial \alpha} \frac{\partial \alpha}{\partial \xi_i}$$
$$= \rho_0 P \frac{\partial \alpha}{\partial \xi_i}$$
$$= \rho_0 P \frac{\partial \alpha}{\partial r_i}$$
$$= \rho_0 \frac{\partial (P\alpha)}{\partial r_i} - \rho_0 \alpha \frac{\partial P}{\partial r_i}.$$

Is this correct? If so, what has happened to the $\rho_0 \frac{\partial(P\alpha)}{\partial r_i}$ term? This will be important for bounded flows.

9. Eq (17): Inner product symbols on first and last terms in brackets.

10. Just below Eq (19): I wonder if you want to reference Sec 4 to point reader here to where the changes required in 2d would occur (e.g., the factor of $1/4 \to 1/3$)?

11. Eq (20): Inner products on LHS and RHS.

12. line 164 and 166: you don't need the vector symbols on the $\vec{p}_i$, since you're referencing by momentum component, right? Also, just to be clear, you don't retain the correction (right-most) term in the equation on line 166 in the tests, right?

13. Eq (23): The $\tilde{H}$ should have a $\tau$ subscript, correct?

14. Eq (23): This will sound like a silly question, but it's important for your discussion of implementation on line 266: to be clear, each vertex is represented in the sum in Eq (23) only once, correct? Also, to be clearer still: there is no restriction on the number of tetrahedra/triangles that may share a given vertex, is there?

15. line 192: Isn't 'reassignment' really just Lagrangian remapping?

16. Eq (26) & (28), et seq: This formulation is the well-known 'barycentric approximation', isn't it? If so, I think this should be stated, and a reference provided.

17. Eq (27): If this is the barycentric approximation, then the condition on the $\tau_i$ should be $1 \geq \tau_i \geq 0$ (including the 1 and 0), and there's also a condition $\sum_i \tau_i = 1$ that should be provided.

18. Eq (29): But this is only a single equation for three $\tau_i$. Do you substitute $\vec{R} = \vec{a}_i, i = 1,3$ and solve the linear system then? Can you please add this? Also, I don't see clearly how solutions to Eq (29) select the tetrahedra from which the fluid particle came. Is the ordering you gave in Eq (27) used somehow to determine this? Normally, the barycentric approximation is used to interpolate within a given tetrahedron, but we have to know already which tetrahedron we're in. Can you please clarify this? [Note: a schematic illustration might help.]

19. line 218: 'the topological structure of the initial and shifted points  ⟨does not⟩ change': But, since you're remapping to an Eulerian grid, why does this matter?

20. line 278: One of the key aspects of your formulation and implementation is the potential for vectorization which will lead to significant performance gains on-node. And, if the vertices are unique, and domain decomposition is done via the individual tetrahedra, it begs the question as to how you handle the data exchange for "coarse-grain" parallelization. These questions certainly interest me, and are critical considerations for dynamical cores. I know that this issue is not the main focus of this paper, but should you perhaps say something here or in a following (sub-)section about performance considerations, even if it's speculative at this point? Depending on what you decide, some mention of performance (potential, or actual numbers) might be added to the abstract, and to the conclusion.

21. Section 4: This is a nice test problem. But the intent should be to validate your numerical formulation and its implementation. Given this, and assuming there is no (quasi-)analytic solution to this test, it seems crucial to compare your solution, at least casually/observationally, with that from a validated numerical solver, unsatisfying as this might be. Perhaps the Editor can make a decision about this, if it's too onerous a thing to do.

22. line 354-356: In the atmosphere, as for other geophysical systems, irreversible mixing is a key element for mesoscale finer scale statistical behavior. This is determined by not only the existence of entropic behavior, but on the manner in which dissipation is handled (e.g., there are significant differences between those that act mainly at low vs. higher wavenumbers). And, whatever the dissipation mechanism, the amount must be accounted for in the energy/entropy. Can the term provided on line 361 ensure this?

**2    Editorial considerations**

Below, the angle brackets, $\langle$XXX$\rangle$, indicate that the text 'XXX' should be added. The strikeout, $\overline{XXX}$ means that the 'XXX' should be removed. The arrow, $\rightarrow$, suggests an emendation.

1. Please double-check the font size requirement; I found it to be too small.

2. line 30: 'to achieve this within $\langle$the$\rangle$ Lagrangian/Hamiltonian approach': add the 'the'.

3. line 36: 'Salmon and Simth 1994' $\rightarrow$'Salmon and Smith 1994'. Please check elsewhere.

4. line 38: 'calculate $\langle$the$\rangle$ state of the atmosphere'

5. line 39: 'one approximately calculates' $\rightarrow$'one calculates approximately'

6. line 41: 'Such an approach seems to be more consistent': More consistent than what? The first approach on line 30?

7. line 43: 'approximation of $\langle$the$\rangle$ corresponding density.'

8. line 47: 'dynamic$\langle$s$\rangle$ of a continuous atmosphere'; add the 's'.

9. line 47: 'a combination of $\langle$the$\rangle$ finite-element method'

10. line 49: 'calculated in a noncanonical coordinates, what leads' $\rightarrow$'calculated in noncanonical coordinates, which leads': remove 'a', and change 'what' to 'which'.

11. line 52: 'of the action in $\langle$a$\rangle$ spatial domain': add the 'a'

12. line 52: 'only assuming a continuous dependence' $\rightarrow$'by assuming only a continuous dependence'

13. line 53: 'one can discretize ⟨the⟩ calculation'

14. line 53: 'not only in spatial but in time domain as well': Awkward. Suggest: 'not only in a spatial domain, but in the time domain as well'

15. line 61: 'that precede ⟨the⟩ transition'

16. line 62: 'with the following ⟨(Eulerian)⟩ equations'. This, to distinguish your Eqs (25) later on.

17. line 67: 'we denoted pressure with ⟨a⟩ capital letter). In  Lagrangian'

18. Eq (3): The first determinant should have an exponent of -1, instead of 1.

19. line 85: 'demonstrate that  Eq. (2)': don't need 'equation', since you have 'Eq.'.

20. "Decorations" like the vector symbol aren't showing up all that well. This may a problem with my printer, or it may be a math font problem (or both). Any way you can check this?

21. line 102: 'is considered  ⟨as⟩ a function'

22. line 139: is 'mode' supposed to be 'model'?

23. line 148: 'accuracy of ⟨the⟩ square of the ratio of the linear size of ⟨the⟩ terahedra'

24. line 152: 'contribution from all  ⟨tetrahedra⟩'

25. line 154: $\vec{p}\,\partial_t\vec{\xi}$ should be $\vec{p}\cdot\partial_t\vec{\xi}$ , right?

26. line 178: 'parenthesis' →'parentheses'. Also, there are two summations, and two sets of parentheses; to which are you referring. I get it, but it's not clear.

27. line 187-188: 'corresponding conservation laws; conservation will be exact; however' →'corresponding conservation laws. In this case conservation will be exact; however'

28. Section 3: This section is a significant fraction of the paper length. If there is a way to **at least** provide subsections, this would likely aid the reader, and look less dense. I would suggest that subsections could be added without any re-structuring in at least 2 places: You might provide a subsection called 'Lagrangian reassignment and local Hamiltonians' starting on line 192 that goes until line 265, or you can separate lines 192-229 into one subsection, and lines 230-265 into another. I suggest that lines 266-278 should logically form its own subsection on implementation.

29. line 193: 'to reassign values of momenta $\vec{p}_k$⟨,⟩ and'

30. line 217: 'existence and uniqueness of such ⟨a⟩ solution follows'

31. Eq (30: place comma after equation.

32. line 239: 'After⟨wards,⟩ the LHS...'

33. line 260: 'the issue of ⟨an⟩ absorbing layer'

34. line 271: 'all the  ⟨tetrahedra⟩ are...'

35. line 275: 'that at a particular area' →'that in a particular region'

36. line 285-286: 'In the 2D case  ⟨tetrahedra⟩ are replaced'

37. line 317: 'The form of ⟨the⟩ Hamiltonian'; 'the action  ⟨does not⟩ change'

38. line 321: '(see Eq. (38), in this case' →'(see Eq. (38); in this case'

39. line 326: 'in ⟨the⟩ vertical, an 200 s in time'

40. line 329: 'and  ⟨a⟩ fourth-order'

41. line 332-333:'accounted  ⟨for⟩. It was found that mass was conserved to  machine precision; conservation of energy  ⟨was not⟩'

42. line 347: 'motion of the  ⟨continuum⟩ atmosphere'

---

## Referee Comment (RC3)

[referee-annotated manuscript omitted]

---

## Author Response (AR1)

**Response to Reviewer 1**

The author is grateful to the anonymous reviewer for drawing attention to (Salmon, 2007) paper which was overlooked. Two additional related papers by R. Salmon: "A general method for conserving energy and potential enstrophy in shallow-water models", J. Atm. Sciences, 64, 515-531, 2007 and "A general method for conserving quantities related to potential vorticity in numerical models", Nonlinearity, 18, R1-16, 2005 will be added to the references list, and the following sentence will be also added to the second paragraph of the Introduction:

"In (Salmon, 2005; 2007) a technique was developed which is based on discretization of a Nambu bracket (a generalization of the Poisson bracket) and allows derivation of spatially-discretized equations of motion which exactly conserve energy, potential enstrophy, mass, and circulation."

The author totally agrees with the reviewer that suggesting development of a dycore for GCMs based on the approach presented in this paper would be at this point premature. Thorough comparisons of how this approach fares against more traditional ones in different practical situations is needed; however, this requires additional research and cannot be accomplished within a single publication.

The comparison of the model developed in (Salmon, 2007) and the model suggested in the paper is definitely of great interest, especially so since they are due to different approaches: approximation of the continuous Hamiltonian equations by a discrete analog on one hand, and approximation of the action of the continuous system by a discrete set of canonical variables on the other. Note also that they consider in fact different systems: shallow water equations and compressible atmosphere. It was mentioned in the concluding Section of (Salmon, 2007) that corresponding generalization for the nonhydrostatic primitive equations would be non-trivial; thus, re-formulation of the approach of this work for shallow-water equations might be easier. However, this requires an additional research and should be done in my view in a separate publication.

**Response to Reviewer 2**

The author is very grateful to the reviewer for his/her review which is really in-depth. The author was very encouraged to know that expert in the field finds the approach advocated in the paper worthwhile. The review is so helpful and goes so far beyond what usual reviews do that, in the case the paper is accepted for publication, I would be happy to acknowledge contribution by the reviewer by name, if he/she is willing to open it.

Practically all Editorial Considerations were accepted. Three exemptions are comments 18, 20, 29. Comment 3: Eq. (3) seems to be correct; comment 20: the arrows denoting vectors look OK in my computer; comment (29): line 217 seems to be OK.

As Editorial comment 28 suggested, two new subsections: 3.1 and 3.2 were introduced.

Below follow detailed answers to the Technical Considerations put forward by the Reviewer.

1. The author agrees that more consistent "dot" notation for the scalar product is preferable. Corresponding modifications will be made throughout the paper, in particular in Eqs. (1,7,8,9,16,17,20,23) and in a few unnumbered equations. The typo in the continuity equation will be fixed.
2. The rotation term in Eq. (7) is a scalar. It becomes obvious when dot is added between vector $\vec{\Omega}$ and the following parenthesis.
3. The reason for absence of the factor $\sum C_{ij}$ in the boundary term is that the divergency term is calculated in the $\vec{r}$ - and not $\vec{a}$ -coordinates:

$$-\int \sum_{i,j=1}^{3} C_{ij} \frac{\partial}{\partial a_j}\left(P\delta\xi_i\right)d\vec{a} = -\int \sum_{i=1}^{3} \frac{\partial}{\partial r_i}\left(P\delta\xi_i\right)d\vec{r} = -\int \left(\vec{N}\cdot d\vec{\xi}\right)Pd\Sigma$$

since from the expression for $C_{ij}$ in line 98 one has

$$\sum_{j=1}^{3} d\vec{a}C_{ij} \frac{\partial}{\partial a_j} = \sum_{j=1}^{3} d\vec{a}\left|\frac{\partial\vec{r}}{\partial\vec{a}}\right|\frac{\partial a_j}{\partial r_i}\frac{\partial}{\partial a_j} = d\vec{r}\frac{\partial}{\partial r_i}$$

4. Dots will be added to indicate corresponding inner products what makes the LHS of (9) manifestly scalar.
5. Integration with respect to time of the term written down by the Reviewer in his/her comment gives:

$$\int d\vec{a}\int_{t_0}^{t_1} dt\partial_t\left[\delta\vec{\xi}\cdot\left(\partial_t\vec{\xi}+\vec{\xi}\times\vec{\Omega}\right)\right] = \int d\vec{a}\left[\delta\vec{\xi}\cdot\left(\partial_t\vec{\xi}+\vec{\xi}\times\vec{\Omega}\right)\right]_{t_0}^{t_1}$$

Variations $\delta\vec{\xi}$ at $t=t_0$ and $t=t_1$ are assumed zero throughout the medium (i.e., for all $\vec{a}$) what is possible since they are arbitrary. As a result, correct equations of motion follow.
6. A hydrostatic state is an exact solution of Eqs. (6), (25) because in this state the potential energy has a minimum; i.e.:

$$\frac{\delta}{\delta\vec{\xi}}\left[\rho_0\left(\vec{a}\right)\left(E\left(\alpha,s_0\left(\vec{a}\right)\right)+g\vec{n}\cdot\vec{\xi}\right)\right] = \frac{\delta}{\delta\vec{\xi}}\rho_0\left(\vec{a}\right)E\left(\alpha,s_0\left(\vec{a}\right)\right)+\rho_0\left(\vec{a}\right)g\vec{n} = 0$$

For this reason, $\vec{\xi}_k = 0$, $\vec{p}_k = 0$ is a (hydrostatic) solution of Eq. (25) for all $k$ (i.e., for all vertices). Thus Eq. (25) allows linearization around the hydrostatic state. No modifications of the Lagrangian or Hamiltonian are necessary.

It is worth mentioning here that in the 2D case considered in Sec. 4, to ensure that the hydrostatic equilibrium is a solution of governing equations, one has to introduce a "calibration" function $\Psi\left(s\right)$. For this reason, treatment of the dynamics in terms of Lagrangian coordinates (in a "hydrodynamic" sense of the word) seems to be preferable.

7. The author apologizes for the typo; the tilde sigh in the unnumbered equation following Eq. (16) should be absent. The following sentence will be added following Eq. (4) for

clarity: "*The "star" index in $L_*$ is introduced to distinguish the total Lagrangian $L_*$ from its (spatial) density $L = dL_* / d\vec{a}$ where $d\vec{a}$ means an element of volume with respect to $\vec{a}$ -coordinates.  The same pertains to the Hamiltonian in the equations below.*"

8.  Calculation of the variation of the internal energy term with respect to $\vec{\xi}$ includes a non-trivial transition between $\vec{a}$ - and $\vec{r}$ - coordinates, what leads to extra factors in the term following the second equality sign in the Reviewer derivation.  The pressure gradient term in the unnumbered equation below (16) follows from Eq. (8) (for points inside the volume the surface term doesn't contribute).

9.  Two corresponding dots will be added.

10. The following sentence will be added after Eq. (19): "*The same equation holds also in 2D case with $1/4$ factor replaced by $1/3$ and summations proceeding from 1 to 3.*"

11. Two corresponding dots will be added.

12. This point does need a clarification. The following sentence will be added after Eq. (20): "*Here and below vector symbols (arrows on top) indicate 3D vectors, and indices correspond to vertices. Dots between vectors denote scalar products with respect to 3D vector coordinates, correspondingly.*"

13. Yes, this is correct.  Index $\tau$ will be added to $\tilde{H}$ in Eq. (23): $\tilde{H} \rightarrow \tilde{H}_\tau$.

14. Yes, this is correct; each vertex in Eq. (23) appears only once and there is no restrictions on the number of tetrahedra sharing a vertex.

15. It seems so. However, not being an expert in modeling atmospheric dynamics, I avoided the term "Lagrangian remapping" not being sure if it assumes more than just setting initial conditions.
    My understanding is that the initial conditions in Eulerian and Lagrangian descriptions are in fact the same: at $t = 0$ to each point in space a velocity vector corresponding to the fluid particle located at this point is assigned. Then either the shift vector of the same fluid particle as a function of time is chosen to represent the evolution, or the velocity vector at the same spatial point.  Since we are using the Lagrangian description (the shift vector $\vec{\xi}$ ) within the time step, to perform a next time step we have to formulate the initial condition at $t = \Delta t$. For the Eulerian description this is just the velocities obtained at the end of the previous timestep. For the Lagrangian description this is not that straightforward and requires figuring out first what particle has arrived to a given vertex at $t = \Delta t$, and then assigning to corresponding vertex the particle's velocity (or momentum). If this is in fact "Lagrangian remapping", I will use this term in the paper.

16. No, $\tau$ s in  Eq. (26) are not barycentric coordinates (in the latter case there should be four of them.*)

17. Since $\tau$ s are not barycentric coordinates, the condition $\sum_i \tau_i = 1$ doesn't hold.

18. Since (28) is a vector equation, we have in fact three equations for three scalar parameters $(\tau_1, \tau_2, \tau_3)$. The reviewer's statement is correct that solution of Eq. (29*) per se* doesn't determine from which tetrahedron the particle came: this is done due to condition (27). The most straightforward approach is to solve linear set (27) for all

tetrahedra containing corresponding vertex, although different plausible "first guesses" could be suggested to save computation time (which is not too big for solution of the third order linear set anyway).

19. What is meant here that the shifts $\vec{\xi}$ should be small as compared to the tetrahedra sizes. Then the tetrahedra do not degenerate within a time step and the field of shifts $\vec{\xi}(\vec{a})$ ensures one-to-one correspondence between the original and shifted positions of fluid particles. As a result, one and only one fluid particle arriving to each vertex can always be found, although it is generally not known in advance form which tetrahedron.

20. My (speculative) vision of the parallelization option is as follows. Each available core handles a set of connected tetrahedra. Data exchange between cores will include only tendencies for the "boundary vertices", i.e. vertices which belong to tetrahedra which are handled by different cores. Since the boundary vertices constitute generally a small share of all vertices, corresponding data fluxes will hopefully be not too intense. I would imagine one central core which receives tendencies for the boundary vertices from all cores and redirects these tendencies to corresponding counterpart cores (note, that due to splitting/recombination of tetrahedra the set of boundary vertices may also vary, and the central core has to track corresponding changes).
Regarding computer realization of the code I would like to mention the following. Essentially all job will be done by a single function which handles a single tetrahedron. This function takes at the input $\left(\vec{\xi}, \vec{p}, \rho_0, s_0\right)$ parameters at the four vertices ( $(3 + 3 + 2) \cdot 4 = 32$ parameters altogether) and at the output provides by calculating corresponding Hamiltonian (given by Eq. (32)) and its derivatives 24 tendencies $\left(\partial_t \vec{\xi}, \partial_t \vec{p}\right)$ at the same vertices. This function is run within a single loop over all tetrahedra. Accumulation of the (weighted by volumes) tendencies at every vertex provides total tendencies for all vertices $\left(\partial_t \vec{\xi}_k, \partial_t \vec{p}_k\right)$. These tendencies represent input for any suitable time-integration scheme.

21. The author agrees that comparison of the numerical results obtained in Sec. 4 vs. established algorithm would be highly desirable. Such undertaking is, however, too hard for a single researcher who doesn't have an access to the existing dynamical cores nor experience in running them. Thus, the only test of validity of the numerical solution in Sec.4 provides conservation of mass. On the other hand, the main purpose of the paper is not to prove correctness of the developed code but to demonstrate feasibility of a code based on the suggested approach including an option of splitting/recombining tetrahedra.

22. This seems to be the case for at least Reynold stresses. Another important factor is heating/cooling of the air which can be taken into when reassignment of the entropy at the vertices is done. This is all pretty much speculative at this stage, though. Still, if a value the approach advocated in this paper is demonstrated in the simplest conservative case, there seems to exist ways to accommodate dissipative processes.

**Response to Reviewer 3**

The author is grateful to the reviewer for the thorough, useful review. The reviewer concisely and absolutely correctly summarized the essence of the paper in the introductory sentence.

Below follow detailed responses to his/her comments.

1. Thea author agrees with the Reviewer that the approach by Gawlik and Gay-Balmaz seems to be the closest to the one used in the current work.  A valid point was made by the Reviewer that this is not correct that "…. non-canonical variations can be avoided in the present manuscript because time is kept continuous". The author agrees; however, the paper didn't mean to claim that.  Such an impression was, apparently, due to the fact that the paragraph mentioning continuous time immediately followed the citation of the Gawlik and Gay-Balmaz work.  To clarify the issue and emphasize the difference between the Gawlik and Gay-Balmaz's work the sentence starting in line 50 is modified as follows:
   "This interesting technique differs significantly from the approach pursued in this paper where the action is calculated in the canonical coordinates and there are no restrictions on the coordinates/momenta variations".

2. The author agrees that the procedure of Lagrangian reassignment aka semi-Lagrangian advection is well known for ages. The following sentence is added to the introductory paragraph of subsection 3.1: "The familiar procedure of the Lagrangian reassignment (aka semi-Lagrangian advection, Lagrangian remapping, etc.) is detailed in this subsection in conjunction with the linear interpolation of the coordinates/momenta employed in this work".

3. The term "forward operator" is used in the paper five times. In two cases its relation to the interpolation is explicitly mentioned.  In the three other cases (lines 8, 46, 353) a qualification "i.e., the mode of interpolation" is added.
   After mentioning "discrete set of parameters" in line 37 the following qualification "i.e., the finite number of degrees of freedom" is added.

4. The author agrees with the Reviewer; "in a broad sense" statement is replaced by "from the standpoint of action minimization".

5. Similar comment (# 21) was made by Reviewer 2.  The author agrees that comparison of the numerical results obtained in Sec. 4 vs. established algorithm would be highly desirable. Such undertaking is, however, too hard for a single researcher who doesn't have an access to the existing dynamical cores nor experience in running them. Thus, the only test of validity of the numerical solution in Sec.4 provides conservation of mass.  On the other hand, the main purpose of the paper is not to prove correctness of the developed code but to demonstrate feasibility of a code based on the suggested approach including an option of splitting/recombining tetrahedra.

6. Investigation of conservation properties although very important *per se* is beyond the scope of this work. The statement in lines 186-189 only emphasizes their dependence on presence of continuous symmetries.

7. Reviewer 2 was interested in certain details of possible realization of the algorithm.  Since corresponding comments take of only a couple of sentences, the author would suggest to leave the sentence in line 274 as is.

8. The phrase "For historical reasons" is removed from the text.

---

## Referee Report (RR1)

This is a second review of the mauscript entitled "Dynamically-optimal models of atmospheric motion" by A. G. Voronovich.

I appreciate the attempts by the author to adddress the previous comments. And, again, I believe the maunscript has merit due to the significant potential advantages of the new scheme, and the–in my experience–novel use of the discretized action for deriving the equations of motion that will be solved numerically, such that these equations obey by construction the conservation properties reflected in the symmetries of the Lagrangian.

That said, in the end, the development provided is of a numerical scheme that solves the conservation laws for geophysical fluids, Over many years, the atmospheric "modeling" community has advanced several types of tests that are regarded as "standard" test cases in regional contexts that validate new numerical schemes. Several are provided below, and include 2d **and** 3d tests. The single 2d test that the author has provided, while reflecting a nontrivial case, does not adequately validate the scheme to the level the community demands, I'm afraid.

Indeed, since the scheme must be implemented on a computer, it is strongly desirable to at least pay hommage to potential performance benefits of the formulation, since good performance must be exhibited if the scheme is to be adopted. In my opinion, this need not be exhaustive at this point, mainly because of the novelty of the scheme, and because this is not the focus of the manuscript at this point in the development. But the author possesses a lot of material to use in offering such a tribute, as mentioned in my previous comments!

Finally, I still have concerns about the Lagrangian remapping ("reassignment"; see below) that is a key aspect of advancing the solution at each vertex or grid point that is a common problem with Lagrangian-Eulerian formulations: how do you ensure uniqueness or non-degeneracy of the solution due to converging flow-lines that might want to cross at a point?

I hope the author will make these changes and resubmit, as I feel that the author's treatment can offer potentially significant value to the geophysical modeling community. I want to thank the author for the kind invitation to join as a co-author, but I must respectfully decline due mainly to conflicts of interest in my capacity as a reviewer. I am, however, willing to offer constructive advice as best I can, as I would like to see this manuscript published if the issues identified can be addressed adequately!

Regarding 'technical issues' below, if one of these derives from my previous list, I'll include the number from that list in parentheses just after the new enumeration. I do not guarantee that the list of editorial comments is as comprehensive this time, but please note these below. The author made a lot of changes that have certainly improved the readability of the manuscript.

**1 Technical considerations**

1. Eq. (17) and Eq. (21): add a dot symbol between $\vec{p}$ and $\partial_t \vec{xi}$

2. (item (8) in previous comments) $\partial_t \vec{p}$ equation just below Eq (16): I certainly see that this equation follows from Eq. (8), but the boundary term is missing entirely here even though it's in Eq. (8)! In addition, as I showed in the previous comment item (8), it should also appear in this equation from straight-forward computation of the variational derivative acting on the pressure gradient term. Also, the lack of this term in this equation makes me wonder if you retained it in your test problem, and, if not, why not? Note that one of the test problems suggested below (the density current test) will likely highlight the inclusion of this term.

3. (items 16-17 of previous comments) Eq. (27) Given that this is not a barycentric approximation, I still don't fully understand where this ordering comes from. Can you please elaborate?

4. (items 16-18 of the previous comments) Around line 228: 'the condition in Eq. (27) selects the appropriate tetrahedron...': But is the selection of the originating tetrahedron unique? What is it about Eq. (27) that guarantees uniqueness? It doesn't appear that anything prohibits two or more particles from having paths that intersect at a vertex simultaneously, so isn't the solution for the upstream properties inherently degenerate? This should be discussed in the manuscript.

5. (relates to item (20) of the previous comments) Before Sec. 4: As noted above, it would helpful to identify either exiting or potential opportunities for performance optimization. The vectorization opportunities mentioned before is one, and the potential to optimize cache utilization is another. Given the manner in which you replace the sums over tetrahedra in Eq. (21) with the sums over unique vertices in Eq (23) suggests to me, perhaps naively, increased utilization of data stored in cache for floating point operations, which would increase so-called "computational intensity". All these things can be measured, and are referred to as "on-node" performance measures. You might consult a couple of recent papers that have considered these issues: Bertagna, et al. 2019 Geosci. Model Dev., 12:1423–1441, and Rosenberg et al. 2023 MWR 151:2521-2540. I don't think you need to measure these things in your test problems, though some reviewers may disagree. But it is important to cite these things as areas of future work to demonstrate that you're considering them even at this early stage.

   Geophysical flows, as you know, are often very high Reynolds number flows, and, therefore, require a lot of grid points to resolve the scale interactions accurately. Thus, it will also likely be important to touch on coarse-grain parallelization achieved by domain-decomposing the grid into subdomains and using the Message Passing Interface (MPI). This will require a little thought, as you should say something on how you will exchange data between subdomains so that you can have each MPI task or process continue to integrate your equations, and whether anything in your formulation will prohibit this (I do not believe that anything prohibits

this, and I believe the domain decomposition of subdomains comprising tetrahedra is well understood). Again, I don't think you need to implement these things at this stage, but to sort of outline the considerations required to achieve good strong scaling, in order for your method to useful as a dynamical core. A few relevant papers on this include the Bertagna et al. reference, as well as Kelly and Giraldo,2012 J. Comput. Phys., 231:7988–8008, and Dennis et al. 2012 J. High Perform. Comput. Appl., 26:74–89.

6. (relates to item (21) of the previous comments) While you have provided a nice *potential* test problem, it cannot be used for validation purposes here because you have not provided a basis for comparison of the results of your scheme. Hence, I believe the editor will require that you provide some community-accepted test problems. You do not need to run a companion dynamical core, since many others have run these problems, so you only need to reproduce these results adequately with your scheme. Since you have formulated your scheme in 3d, it will seem oddly incomplete not to validate in 3d as well, so this should be done, in my opinion. Unfortunately, this will likely require at least coarse-grain parallelization in order to achieve. The above references contain many of the more popular ones, but I will distill them with references here:

   (a) 3D mountain wave:
       - Lock et al. 2012 MWR 140:411-424
       - Melvin et al. 2019 Quart. J. Roy. Meteor. Soc., 145:2835–2853
   (b) 2d density current, testing critical boundary effects, and multi-scale interaction:
       - Straka, et al. 1993 Int. J. Numer. Methods Fluids, 17:1–22
   (c) Linear gravity waves:
       - Baldauf, M., and S. Brdar 2013 J. Roy. Meteor. Soc., 139:1977–1989
       - Rosenberg et al. 2023 MWR 151:2521-2540
   (d) 3D rising thermal, testing basic coherent buoyancy within a 3d volume:
       - Shapiro, A., and K. M. Kanak 2002 J. Atmos. Sci., 59:2253–2269;
       - There are many versions of this type of test in the other references above.

**2 Editorial considerations**

Below, the angle brackets, ⟨XXX⟩, indicate that the text 'XXX' should be added. The strikeout,  means that the 'XXX' should be removed. The arrow, →, suggests an emendation.

1. line 22: 'The approaches  ⟨that⟩ start'

2. line 23: 'equations of motion and ⟨are⟩ pursued'

3. line 36: 'by modifying ⟨the⟩ corresponding'

4. line 41: 'one calculates approximately ⟨the⟩ action density'

5. line 43: 'From its standpoint' →'From this standpoint'

6. line 48: 'ODE' →'ODEs'

7. line 138: 'and  ⟨inserting⟩ from the result'

8. line 171: Reference to Eq. (29) should be to Eq. (19), right?

---

## Referee Report (RR2)

This is a (final) review of the mauscript entitled "Dynamically-optimal models of atmospheric motion" by A. G. Voronovich.

The author has addressed most of my concerns in his response, which I appreciate. I believe that, with now minor changes, the manuscript can accepted for publication.

First, and not in order of the author's responses, the reason I emphasized both "standard" test cases, and to paying homage to potential performance benefits, is because the very first line of the abstract says that the paper is a "derivation of a dynamical core...". This leads many readers down a very well-worn path of expectations. If this alone were changed to "A derivation of the dynamical equations for the dry atmosphere...", I think the concerns regarding the test problems and performance would be alleviated. Even in this case, however, it would still be helpful to point out in the conclusion potential additional development, *including* optimization benefits that can be examined, and standard tests for a dynamical core that could be run. I would try to withhold stating anything about dynamical cores until the conclusion, in fact (although the comment in the Introduction about the method being of interest to developers of dycores is ok, IMO).

I want to thank the author for the detailed responses to my other comments. Looking at my previous comment 2 on the missing boundary term, I fully see my misconception, and no further explanation is required in the manuscript. The author's explanation for the affine transformation is also appreciated; it is, in fact, the same as one would use in the barycentric approximation.

Regarding the author's response #4 to my question(s), however, I still have some questions, but this response engenders the only modification that I recommend. First, on line 345, you indicated that $\Delta t = 0.25$, but you do not state how you arrived at this. But I believe this is very much related to your response # 4 where you state that the time step in essence ensures uniqueness and well-posedness. You need to state how you chose this time step. But you might effectively include this in a modified discussion of uniqueness and nondeneracy that you started on line 228: It seems to me that by limiting the time step, you are limiting the distance over which the fluid can flow to each $\vec{a}_i$, which is essentially what up-winding attempts to capture in the Eulerian context as it constrains the amount of fluid that flows over one time step. The time step would be governed by the characteristic width of a tetrahedron and the maximum wave speed in the problem (here, the sound speed). That said, I would recommend replacing the sentence on line 228 beginning "The existence and uniqueness of such a solution ..." with a new paragraph that contains essentially your response #4, something like the following (but please read carefully as there are some changes):

*The existence and uniqueness (non-degeneracy) of solutions to Eq. (29) may be explained as follows. Let us consider the trajectories of fluid particles. The fluid particle that at $t = 0$ was located at the vertex with coordinates $\vec{a}_i$ at a later moment of time $t$ will be located at a point $\vec{r}_i(t) = \vec{a}_i + \vec{\xi}_i$. Points $\vec{r}_i$ form vertices of a shifted tetrahedron onto which the initial tetrahedron is mapped. Note, that shifts $\vec{\xi}$ of the fluid particles inside a tetrahedron are assumed to be linear*

*functions of the shifts of the fluid particles located at the vertices of the tetrahedron: this is our basic (linear) approximation of the forward operator (see the first paragraph of Sec. 3). Thus, the initial tetrahedron with the vertices at the points $\vec{a}_i$ linearly (more precisely, affinely) mapped onto a shifted tetrahedron with vertices at $\vec{r}_i$; the transformation is linear regardless of the trajectories of the fluid particles at the vertices $\vec{a}_i$ being linear or curved. In particular, faces and edges of the initial tetrahedron are mapped onto corresponding faces and edges of the shifted tetrahedron. Since shifts of the internal points of the tetrahedron are linear functions of $\vec{a}_i$ the Jacobian of the linear transformation of the initial tetrahedron within it is constant (the constants for different tetrahedrons are, of course, also different, and they depend on time t). Thus, piecewise linearity of the forward operator ensures that the mapping of the whole initial volume onto the shifted volume is also piecewise linear, and the mapping is one-to-one provided neither tetrahedron in the course of evolution degenerates (i.e. tetrahedra volumes never become zero). The latter is achieved by adopting a Courant-limited timestep based on the fastest wave-mode that the equations admit (here, the sound speed (correct??)), which also ensures that time integration errors remain small. Non-degeneracy of the initial tetrahedrons can be checked easily, since trajectories of the fluid particles at the vertices are calculated in the course of numerical integration.*

On line 345, you can then just point to this discussion when selecting a time step.

Finally, on line 342: 'periodic in horizontal direction' →'periodic in the horizontal direction'.

---

## Author Response (AR2)

**Reviewer 1.**

1. *I still have substantial concerns about the fundamentel approach. Treating a non-canonical system in canonical coordinates is non-trivial and requires one to pass to Clebsch variables (cf. J. E. Marsden and A. Weinstein. Coadjoint orbits, vortices, and Clebsch variables for incompressible fluids. Physica D: Nonlinear Phenomena, 7(1-3):305–323, may 1983.). In the non-canonical setting, the work by Gawlik and Gay-Balmaz (and earlier work by Gawlik et al, Pavlov et al.) required considerable technical complexity (e.g. in terms of the Hodge star operator) to make the non-canonical system well defined in the discrete setting. The manuscript (and potentially a longer reply to this reviewer) needs to explain why a canonical treatment is possible although one deals with a non-canonical system/description.*

Response

Under a "non-canonical system" the Reviewer, probably, had in mind a Hamiltonian system represented in non-canonical coordinates. Judging from the title, the paper by Marsden and Weinstein mentioned by the Reviewer considers an incompressible fluid, where the Lagrangian description of fluid dynamics provide canonical coordinates only after imposing restrictions. This leads to introduction of Clebsch variables. However, in the case of a compressible fluid considered in the paper the Lagrangian displacements and the products of density and corresponding Lagrangian velocities provides a canonical description. Canonical coordinates are not, of course, unique, and Clebsch variables can be introduced in the compressible case as well, and the 2D numerical example considered in the paper in fact does use them.

I would like to emphasize that the evolution due to canonical dynamics based on the Lagrangian coordinates is used in the paper within each time step; before next time step the Lagrangian reassignment (aka semi-Lagrangian advection, Lagrangian remapping, etc.) has to be applied.

2. *The manuscript would still substantially benefit from a deeper and more numerical results (see my previous review).*

Response

Although the author in principle agrees with this comment, I would like to emphasize that the point of this paper is not a development of a competitive numerical algorithm but rather a demonstration of a feasibility of building an algorithm based on the least action principle. The author believes that the numerical example presented in the paper basically fulfils this task, demonstrating also a capability of a refinement of an unstructured grid in the course

of calculation and account for an orography, which are claimed to be advantages of the suggested approach.

Still, consideration of a well-established example and comparison of the results would, of course, make the paper much more convincing. This is, unfortunately, unrealizable for the author for a practical reason: I am planning retirement in a few months, and corresponding effort would take much longer. Thus, if consideration of a well-established example deemed to be necessary, the paper should be apparently rejected.

**Reviewer 2.**

The author is very glad to know that the approach considered in the paper is found of interest by an expert in the field, and he is very grateful to the Reviewer for his/her encouraging remarks.

In the introductory part of the review among other comment three important questions were raised: (a) a necessity of considering a "standard" test case (a similar concern was expressed by the Reviewer 1 above; (b) the issue of uniqueness/non-degeneracy of the Lagrangian remapping; (c) consideration of the potential performance benefits of corresponding computer realization of the approach. These issues will be addressed when replying to "Technical considerations" below.

1. The dot symbol was added in Eqs. (17) and (21).

2. A contribution from the boundary term in the second Hamiltonian equation written out after eqs. (16) is absent because this equation corresponds only to the points lying inside the volume (in a more technical terms: since $\delta\vec{\xi}$ is an arbitrary function of coordinates $\vec{a}$, one can set it equal to $\delta\vec{\xi}\delta(\vec{a}-\vec{a}_0)$ where a point $\vec{a}_0$ lies within the volume. The boundary integral in this case turns to zero. The resulting volume integral of the variation of the Hamiltonian density $\delta H$ over the whole volume is by definition the variational derivative $\delta H/\delta\vec{\xi}$ at the point $\vec{a}_0$ times $\delta\vec{\xi}$). The contributions from the first (boundary) term in the RHS of (8) contributes only to the points belonging to the boundary of the volume begetting corresponding boundary conditions).

As far as numerical example goes, the boundary conditions correspond to zero mass flux at the top and bottom points of the atmosphere, and the motion was assumed periodic in horizontal direction (i.e., along $x$-axis). Corresponding clarifying sentence was added after the old line 340.

3. The simplest way to make sure that conditions (27) select inner points of a tetrahedron is to make an affine transformation of the $a$-space mapping the vertex $\vec{a}_4$ onto the origin, vertex $\vec{a}_1$ onto a unit along _x-axis (i.e. into the point $\vec{e}_x = (1,0,0)$, vertex $\vec{a}_2$ onto the point $\vec{e}_y = (0,1,0)$ and vertex $\vec{a}_3$ onto the point $\vec{e}_z = (0,0,1)$ of a new Cartesian coordinates. Then (26) becomes: $\vec{r} = \vec{e}_x \tau_1 + (\vec{e}_y - \vec{e}_x)\tau_2 + (\vec{e}_z - \vec{e}_y)\tau_3$, whence $x = \tau_1 - \tau_2$, $y = \tau_2 - \tau_3$, $z = \tau_3$. In the new coordinates the internal points of the tetrahedron are selected by the conditions $x > 0, y > 0, z > 0, x + y + z < 1$ which in fact coincide with (27).

4. To consider the issue of uniqueness/non-degeneracy of the mapping (29), let us consider trajectories of fluid particles. The fluid particle that at $t = 0$ was located at the vertex with coordinates $\vec{a}_i$ at a later moment of time $t$ will be located at a point $\vec{r}_i(t) = \vec{a}_i + \vec{\xi}_i$. Points $\vec{r}_i$ form vertices of a shifted tetrahedron onto which the initial tetrahedron is mapped. Note, that shifts $\vec{\xi}$ of the fluid particles inside a tetrahedron are assumed to be linear functions of the shifts of the fluid particles located at the vertices of the tetrahedron: this is our basic (linear) approximation of the forward operator (see the first paragraph of Sec. 3). Thus, the initial tetrahedron with the vertices at the points $\vec{a}_i$ is linearly (more precisely affinely) mapped onto a shifted tetrahedron with vertices at $\vec{r}_i$; the transformation is linear regardless of the trajectories of the fluid particles at the vertices $\vec{a}_i$ being linear or curved. In particular, faces and edges of the initial tetrahedron are mapped onto corresponding faces and edges of the shifted tetrahedron. Since shifts of the internal points of the tetrahedron are linear functions of $\vec{a}_i$, the Jacobian of the linear transformation of the initial tetrahedron within it is constant (the constants for different tetrahedrons are, of course, also different, and they depend on time $t$). Thus, piecewise linearity of the forward operator ensures that the mapping of the whole initial volume onto the shifted volume is also piecewise linear, and the mapping is one-to-one provided neither tetrahedron in the course of evolution degenerates (i.e. tetrahedra volumes never turn to zero). The latter is warranted for sufficiently small $t$, since at $t = 0$ the transformation is identical. A time step has to be sufficiently small anyway, since the shifted tetrahedron after a time step should not deviate too far from the initial one; otherwise numerical integration will be in error. Non-degeneracy of the initial tetrahedrons can be easily checked since trajectories of the fluid particles at the vertices are calculated in the course of numerical integration. Note also that the degeneration of the initial tetrahedron means that density inside it becomes infinite; if time steps of numerical integration are selected correctly, this could never happen.

5. The interesting issues raised by the Reviewer at comment 5 go well beyond my area of expertise. On the other hand, they are of primary importance if the approach under consideration is going to be implemented. Although I quite understand the reason the Reviewer declined my invitation to co-author this paper, this is disappointing to me, and the invitation is still open.

6. The desirability of consideration of established test cases was also raised by the other reviewer. Below I copy my response.

   Although the author in principle agrees with this comment, I would like to emphasize that the point of this paper is not a development of a competitive numerical algorithm but rather a demonstration of a feasibility of building an algorithm based on the least action principle. The author believes that the numerical example presented in the paper basically fulfils this task, demonstrating also a capability of a refinement of an unstructured grid in the course of calculation and account for an orography, which are claimed to be advantages of the suggested approach.

   Still, consideration of a well-established example and comparison of the results would, of course, make the paper much more convincing. This is, unfortunately, unrealizable for the author for the following practical reason: I am planning retirement in a few months, and corresponding effort would take much longer. Thus, if consideration of a well-established example deemed to be necessary, the paper should be apparently rejected.

   All suggestions listed in "Editorial considerations" are taken into account in the revised manuscript.

---

## Author Response (AR3)

Dear Prof. Toth,

All your and the reviewer's suggestions make perfect sense and all of them were implemented in the revised manuscript. In particular (line numbers below correspond to the revised manuscript):

*1.Modify the first sentence in the Abstract, as suggested by the Reviewer, by changing "dynamical core" to "dynamical equations".*

Done: see line 5.

*2. Indicate in the Conclusions that possible future work includes standard dynamical core tests and code optimization. You would want to mention "dynamical core" only in this, last section.*

The following sentence was added to the Conclusion section (line 394):

"Future work should consider standard tests for a dynamical core and possibilities for code optimization."

*3. Consider adopting the text related to the length of time steps suggested by the Reviewer.*

According to the reviewer suggestion the following text was added to the sentence regarding selection of the time step (line 361):

"(see also the comment on uniqueness of mapping after Eq. (29))."

This text refers to the additional passage concerning uniqueness of the global mapping due a time step (see # 5 below) and it was added according to the reviewer suggestion between lines 229-244.

*4. Consider including an acknowledgement regarding the constructive comments we received from the reviewers in the Acknowledgements section.*

The following sentences were added in lines 401-403:

"The author appreciates constructive comments made by the reviewers. The author is particularly indebted to one of the reviewers whose insights and thorough examination of details helped to significantly improve this work."

5. Following the reviewer suggestion the following passage which is in fact my response to the reviewer (# 4) at the previous round of reviews was added to the text between lines 229-244:

"The existence and uniqueness (non-degeneracy) of solutions to Eq. (29) may be explained as follows. Let us consider the trajectories of fluid particles. The fluid particle that at $t = 0$ was located at the vertex with coordinates $\vec{a}_i$ at a later moment of time $t$ will be located at a point $\vec{r}_i(t) = \vec{a}_i + \vec{\xi}_i$. Points $\vec{r}_i$ form vertices of a shifted tetrahedron onto which the initial tetrahedron is mapped. Note, that shifts $\vec{\xi}$ of the fluid particles inside a tetrahedron are assumed to be linear functions of the shifts of the fluid particles located at the vertices of the tetrahedron: this is our basic (linear) approximation of the forward operator (see the first paragraph of Sec. 3). Thus, the initial tetrahedron with the vertices at the points $\vec{a}_i$ linearly (more precisely, affinely) mapped onto a shifted tetrahedron with vertices at $\vec{r}_i$; the transformation is linear regardless of the trajectories of the fluid particles which started from the vertices $\vec{a}_i$ being linear or curved. In particular, faces and edges of the initial tetrahedron are mapped onto corresponding faces and edges of the shifted tetrahedron. Since shifts of the internal points of the tetrahedron are linear functions of coordinates the Jacobian of the linear transformation of the initial tetrahedron within it is constant (the constants for different tetrahedrons are, of course, also different, and they depend on time $t$ ). Thus, piecewise linearity of the forward operator ensures that the mapping of the whole initial volume onto the shifted volume is also piecewise linear, and the mapping is one-to-one provided neither tetrahedron in the course of evolution degenerates (i.e. tetrahedra volumes never become zero).

The latter is achieved by adopting a Courant-limited timestep based on the fastest wave-mode that the equations admit (here, the sound speed), which also ensures that time integration errors remain small. Non-degeneracy of the initial tetrahedrons can be checked easily, since trajectories of the fluid particles at the vertices are calculated in the course of numerical integration."

6. In addition two typos (lines 150 and 357) were fixed.

Respectfully,

Alexander Voronovich